# Controlling Selenization Equilibrium Enables High-Quality Kesterite Absorbers for Efficient Solar Cells

Xiao Xu[1,2,5], Jiazheng Zhou[1,2,5], Kang Yin[1,2], Jinlin Wang[1,2], Licheng Lou[1,2], Menghan Jiao[1,2], Bowen Zhang[1,2], Dongmei Li [1,2,3], Jiangjian Shi [1] ✉, Huijue Wu[1], Yanhong Luo [1,2,3] ✉ & Qingbo Meng [1,3,4] ✉

Kesterite $Cu_2ZnSn(S, Se)_4$ is considered one of the most competitive photovoltaic materials due to its earth-abundant and nontoxic constituent elements, environmental friendliness, and high stability. However, the preparation of high-quality Kesterite absorbers for photovoltaics is still challenging for the uncontrollability and complexity of selenization reactions between metal element precursors and selenium. In this study, we propose a solid-liquid/solid-gas (solid precursor and liquid/vapor Se) synergistic reaction strategy to precisely control the selenization process. By pre-depositing excess liquid selenium, we provide the high chemical potential of selenium to facilitate the direct and rapid formation of the Kesterite phase. The further optimization of selenium condensation and subsequent volatilization enables the efficient removal of organic compounds and thus improves charge transport in the absorber film. As a result, we achieve high-performance Kesterite solar cells with total-area efficiency of 13.6% (certified at 13.44%) and 1.09 $cm^2$-area efficiency of 12.0% (certified at 12.1%).

Kesterite $Cu_2ZnSn(S, Se)_4$ (CZTSSe) is considered one of the most promising materials for thin-film solar cells due to the advantages including high light absorption, composing of earth-abundant and nontoxic elements, adjustable band gap, environmental friendliness, and high stability[1–7]. Particularly, Kesterite solar cells fabricated using the solution route offer the advantages of easier mass production and reduced manufacturing costs[8]. Over the years, there have been significant advancements in the solution-processed Kesterite solar cells, transitioning from highly toxic hydrazine to environmentally friendly green solvents[9–13]. These developments have contributed to an increase in the record efficiency from 12.6% to 13.0%[14,15]. However, there still exists a considerable disparity between the current efficiency and the theoretical limit (~32%). This difference primarily stems from the high loss of open-circuit voltage, which can be

attributed to the poor crystal quality and the presence of various types of defects[16–20].

The preparation of high-quality Kesterite crystals poses several challenges due to their diverse constituent elements, narrow phase diagram, and complex crystallization processes[21–23]. Firstly, from a chemical reaction standpoint, the crystallization process of CZTSSe relies on solid-gas chemical reactions between the precursor film and selenium in a high-temperature environment. These reactions are influenced by many factors such as the initial selenium content, concentration and uniformity of selenium vapor, reaction temperature, and more[24,25]. Secondly, from a thermodynamic perspective, the reaction between the Cu-Zn-Sn precursor and selenium easily gives rise to the formation of binary or ternary phases with lower Gibbs free energy[22]. These phases can introduce a significant number of defects in

[1]Beijing National Laboratory for Condensed Matter Physics, Institute of Physics, Chinese Academy of Sciences (CAS), Beijing 100190, China. [2]School of Physical Sciences, University of Chinese Academy of Sciences, Beijing 100049, China. [3]Songshan Lake Materials Laboratory, Dongguan, Guangdong 523808, China. [4]Center of Materials Science and Optoelectronics Engineering, University of Chinese Academy of Sciences, Beijing 100049, China. [5]These authors contributed equally: Xiao Xu, Jiazheng Zhou. ✉e-mail: shij@iphy.ac.cn; yhluo@iphy.ac.cn; qbmeng@iphy.ac.cn

the final film[26,27]. Thirdly, from a kinetic point of view, each cation exhibits different diffusion and volatilization rates during crystal growth, leading to the element segregation or loss in the absorber[28–30]. Consequently, numerous reaction conditions during the crystallization process, including precursor composition, reaction temperature, reaction time, reaction atmosphere, selenium concentration, and uniformity of reactants, have an impact on the crystal quality of the absorber[21,31,32]. Precise control and optimization of these reaction parameters are crucial for achieving a high-quality absorber.

The key areas for improving Kesterite crystal growth involve designing the precursor composition and optimizing the selenization reaction process. Regarding precursor regulation, certain approaches have been employed to suppress the segregation of secondary phases and Sn-related defects, such as altering the valence state of precursor cations[16], improving the local chemical environment[26], and cation doping[33,34]. Furthermore, appropriate selenium into the precursor by dissolving or evaporation deposition has been introduced to alleviate the selenium-deficient atmosphere, resulting in a reduction in Se-related defect[24,35]. Comparatively, there has been relatively less focus on optimizing the selenization reaction process and related technology route. In particular, the widely used single-zone graphite-box selenization method has several limitations. Firstly, as a narrow and enclosed space, the graphite box does not allow for independent control of reaction temperature, vapor concentration, and their evolutions[25]. Secondly, due to the volatile nature of selenium and its strong penetration into graphite, selenium deficiency during crystal growth/ripening stages remains an issue, even with increased initial selenium content[36]. Thirdly, the transport of selenium vapor in the graphite box primarily relies on spontaneous diffusion, which hampers the rapid establishment of a uniform selenium environment over large areas and restricts its application in fabricating large-size solar cells. Therefore, there is an urgent need to develop new

technologies that can precisely and synergistically control the various selenization reaction parameters.

In this study, we have introduced a solid-liquid/solid-gas synergistic reaction (SLSG) strategy within a dual-temperature zone scheme to fabricate high-quality Kesterite absorbers. Our approach involves pre-depositing a sufficient amount of liquid selenium onto the precursor film to facilitate liquid-phase assisted phase evolution and crystal growth and subsequent synergistic control of selenium volatilization to balance the film crystallization and organics removals. The key benefits of this strategy can be summarized as follows. Firstly, the presence of liquid selenium provides a high chemical potential, enabling a faster direct formation of Kesterite phase during the initial selenization stage. Secondly, the high concentration of selenium helps stabilize the valence state of the elements and reduces chemical composition variations within the absorber material. These advantages finally enable the fabrication of defect-less and compact Kesterite absorbers. As a result, we have achieved high-performance solar cells with a total-area efficiency of 13.6% (certified at 13.44%) and large-area devices with efficiency of 12.0% (certified at 12.1%) over an area of 1 cm².

## Results
### Key issues in realizing high-quality Kesterite absorbers
Figure 1a schematically illustrates the phase diagram of Kesterite $Cu_2ZnSnSe_4$ (CZTSe) and its secondary phases. It is evident that the phase evolution process is highly dependent on the concentration of selenium vapor and the reaction temperature[22]. When the Se concentration and temperature are low, the preferential formation of $Cu_2Se$, ZnSe, $Cu_2SnSe_3$, and SnSe is favored, while the direct formation of the Kesterite phase necessitates higher concentrations of Se vapor and elevated temperatures. However, achieving a uniform and sufficient Se atmosphere, especially during the initial selenization stages, is challenging in experiment. To investigate this issue, we initially

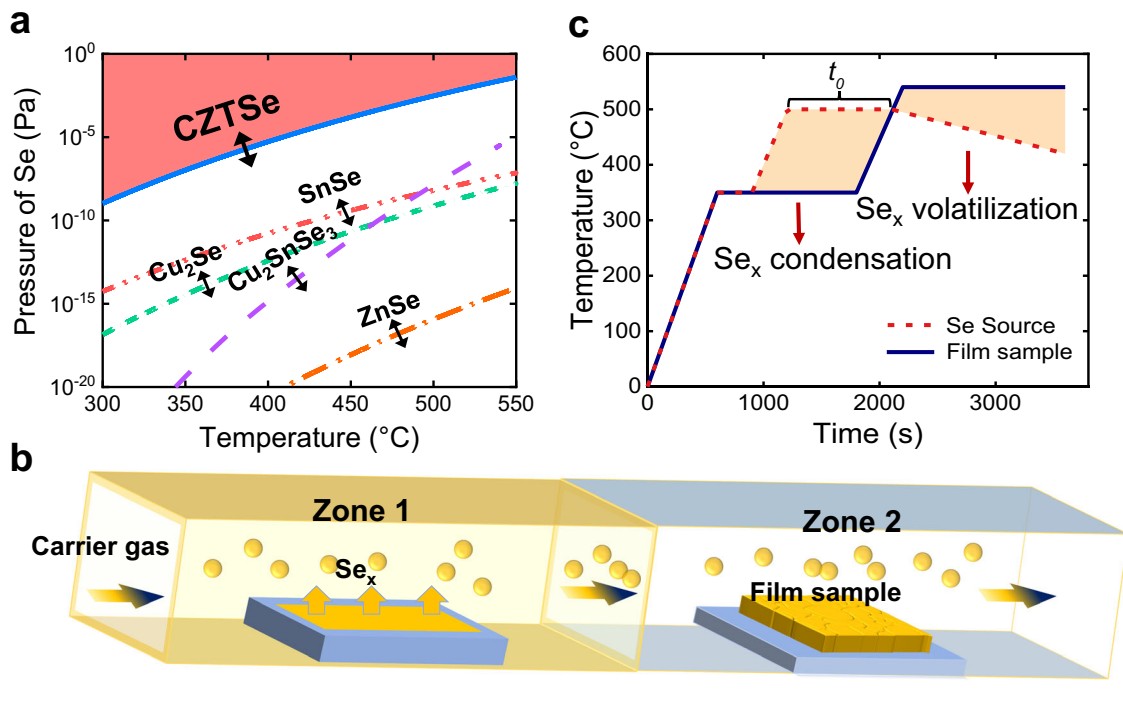

**Fig. 1 | Solid-liquid/solid-gas synergetic reaction strategy. a** Schematic diagram of Se vapor-temperature equilibrium for the formation of CZTSSe and related secondary phases[22,29]. **b** Schematic diagram of the dual-zone selenization system. The $Se_x$ vapor generated in Zone 1 is transported to Zone 2 via carrier gas and subsequently reacts with the precursor film. **c** Schematic diagram of the typical time-dependent temperature evolutions of the precursor film and Se source in this work. The control of Se condensation and volatilization is realized through regulating the temperature difference between these two zones and the preheating duration ($t_0$).

simulated the behaviors of Se volatilization and spatial diffusion in a single-zone graphite box. The results (Supplementary Fig. 1) indicate that the distribution of Se vapor within the graphite box is non-uniform and exhibits a relatively low rate of concentration increase. Additionally, in the confined space of the graphite box, the reaction factors, such as reaction temperature, vapor concentration, and initial amounts of Se reactants, strongly influence one another. Consequently, utilizing the graphite-box selenization technology route may result in multi-step phase evolution, making the occurrence of induced defects almost inevitable.

### Solid-liquid/solid-gas synergetic reaction strategy

To address the issue of multi-parameter coupling, we have adopted a decoupling strategy through space-time separation. Our designed dual-temperature zone selenization approach is depicted in Fig. 1b & Supplementary Fig. 2. In this system, the precursor film and the selenium source are spatially separated, allowing for independent control of their respective heating programs. The molecular selenium is transported from the selenium source zone to the precursor film zone using a carrier gas. By separating the reactants (precursor film and selenium source) and controlling their heating programs independently, we are able to explore a wider range of reaction pathways.

Inspired by the liquid-phase epitaxy techniques employed in Si, SiC, and III-V semiconductors[37–39], here we have chosen liquid-phase selenium to achieve significantly higher molecular Se concentrations compared to Se vapor during the selenization initial stage. The introduction of liquid Se is achieved by establishing a temperature difference between the selenium source and the precursor film zones. Initially, the Se source is heated to a high temperature, causing gaseous Se to be transported to the precursor film with the aid of carrier gas. As Zone 2 maintains a much lower temperature, the supersaturated Se vapor condenses onto the surface of the precursor film, forming a liquid state. Subsequently, as the temperature of the precursor film is increased, the liquid Se reacts directly with the solid precursors. The amount of liquid Se present is directly proportional to the temperature difference and the duration of the temperature gradient. It is important to note that to sustain the presence of liquid Se on the film, continuous and appropriate Se vapor transport from the Se source must be maintained during the high-temperature reaction stage.

### Synergetic optimization in nucleation and growth stage

Specifically, we maintained a constant temperature difference while varying the duration of the preheating stage (indicated as $t_0$ in Fig. 1c) to determine the quantity of liquid Se present in the precursor film. Furthermore, we compared the selenization processes involving solid-gas reactions with those involving solid-liquid/solid-gas synergetic reactions. For clarity, we defined the condition $t_0 = 0$ s as pure solid-gas (SG) selenization and the condition $t_0 = 300$ s as the solid-liquid/solid-gas synergetic selenization (SLSG). The temperature profiles for these two selenization modes are depicted schematically in Supplementary Fig. 3. In our experiment, 10% Ag alloying was used in the precursor, also to improve the absorber quality according to previous literatures (the final Ag-alloyed CZTSSe is abbreviated as ACZTSSe)[4–6,40,41]. Based on the optical microscope images (Supplementary Fig. 4), no liquid-phase Se was observed on the precursor surface under the SG mode. In contrast, in the SLSG mode, the precursor was entirely covered by liquid-phase Se. Top-view scanning electron microscopy (SEM) images of semi-selenized films were captured by interrupting the selenization process after the precursor film had been held at 540 °C for 200 s, as shown in Fig. 2a, d and Supplementary Fig. 5 for the SG and SLSG samples, respectively. In the SLSG sample, a significant amount of cooled liquid Se was observed on the film surface, indicating the successful introduction of a liquid-phase Se-assisted growth process. The presence of liquid Se on the film surface was also confirmed by in-situ photographs taken during the selenization process (Supplementary Fig. 6).

We investigated the impact of liquid Se on the phase formation processes using Raman spectroscopy and X-ray diffraction (XRD) characterization. For this study, we sampled the films at intermediate stages by interrupting the selenization process when the temperature of the precursor films reached 400 and 500 °C, respectively. In Fig. 2b, c, we observed that when selenization followed the conventional SG reaction route, although the Kesterite phase was present, secondary phases such as $Cu_xSe$, $Cu_2SnSe_3$, and SnSe, indicated by the Raman peaks at 265, 180, and 130 cm$^{-1}$[42,43], were also clearly visible. The XRD patterns also revealed a diffraction peak corresponding to the $Cu_xSe$ phase (Supplementary Fig. 7)[44,45]. In contrast, in the SLSG route, these secondary phases were effectively eliminated during the intermediate selenization process, as depicted in Fig. 2e, f. Notably, even when the liquid Se was only partially introduced, the suppression of secondary phases was still apparent (Supplementary Fig. 8). These results indicate that the high molecular Se concentration induced by the liquid phase facilitated the direct and rapid formation of the Kesterite phase.

We conducted a further comparison of the chemical state and composition of these two samples using X-ray photoelectron spectra (XPS) and energy dispersive X-Ray fluorescence (XRF-EDX) spectra. In the SG sample, the Sn $3d^{5/2}$ peak was observed at 485.9 eV with an asymmetric shape, as shown in Fig. 2g. The main component of this peak was fitted to be at 485.8 eV, corresponding to the $Sn^{2+}$ cation in SnSe. In contrast, in the SLSG sample, the main component of the Sn $3d^{5/2}$ peak was located at 486.4 eV, indicating the dominance of $Sn^{4+}$ cations (Fig. 2h)[26,46]. The appearance of the $Sn^{2+}$ cation can be attributed to several factors. Firstly, the insufficient Se in the SG sample results in a Se-deficient condition on the absorber film surface, causing an accumulation of electrons around Sn and leading to the formation of $Sn^{2+}$. Secondly, the lack of Se causes some of the Sn-S precursor to transform into SnSe, affecting the Sn valence after its transformation into ACZTSSe. Thirdly, the Se deficiency can reduce the coordination number of Sn atoms, making the material instable at high temperatures, resulting in the formation of SnSe decomposition products[22,29]. Overall, the presence of $Sn^{2+}$ cations is a direct consequence of the insufficient Se in the SG route. The volatility of SnSe and the inadequate coordination of Sn can lead to high Sn loss at elevated temperatures. This result is supported by our XRF characterization and XPS intensity analysis, as shown in Fig. 2i and Supplementary Fig. 9. The loss of Sn or the Se-deficient atmosphere can induce Sn and Se vacancies[29], which are known to be deep defects in this material[17,21]. In the SLSG sample, the high Se concentration effectively suppresses the Sn loss, thereby helping to reduce these deep defects[47]. We also observed that liquid Se improves the nucleation and morphology evolution of the selenized film, resulting in a more compact grain surface (Supplementary Fig. 10). All of the aforementioned characterizations provide evidence that the solid-liquid reaction is advantageous for achieving high-quality Kesterite crystals, as illustrated schematically in Supplementary Fig. 11. In addition to the Se itself, it was reported that Cu-Se compound could also perform as flux to assist the growth of chalcogenides due to its high reaction activity with other metal elements and the high mobility of Cu cations[30,48]. In our samples, high Se concentration would facilitate the melting of Cu-Se alloying to more effectively assist the crystal growth.

### Synergetic optimization in the ripening stage

In environment-friendly solution routes, achieving a balance between crystal ripening and removal of organics within the precursor film presents another challenge. Similar to the phase evolution and nucleation processes mentioned earlier, these two processes are also significantly influenced by the Se atmosphere. In our experiments, we observed that although better Kesterite ACZTSSe crystals were obtained using the SLSG route, the resulting solar cells exhibited significantly lower power conversion efficiency (PCE) compared to the SG route (Fig. 3a). Specifically, there was a notable decrease in the short-circuit current density ($J_{SC}$) and fill factor (FF). This indicates poor

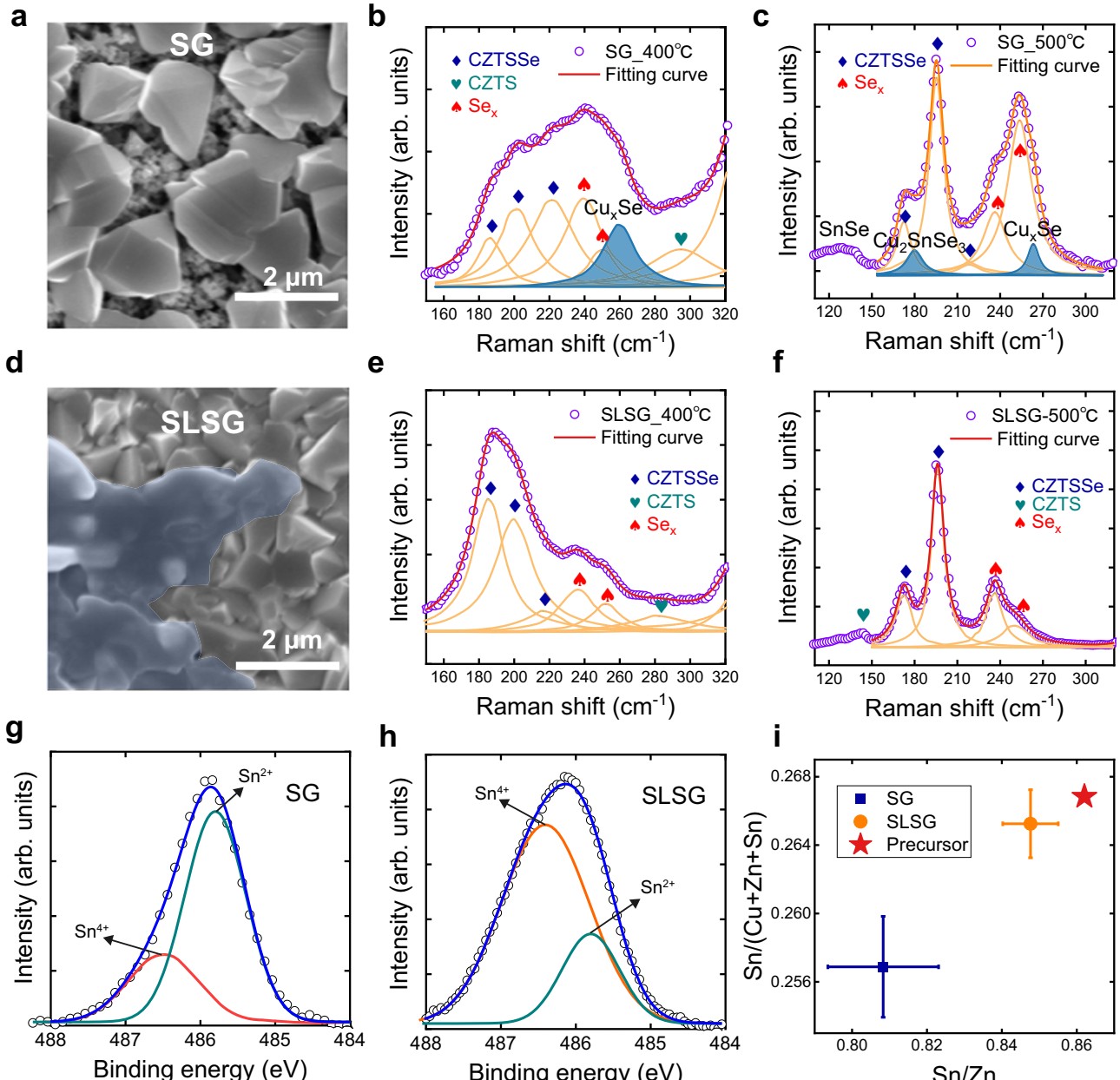

**Fig. 2 | Characterization of solid-gas (SG) and solid-liquid/solid-gas synergetic selenization (SLSG) films.** Top-view SEM images and Raman spectra of selenized films based on SG and SLSG reaction routes (**a**–**c**: SG sample, **d**–**f**: SLSG sample). The bule region in the SEM image is determined to be the cooled liquid Se according to the element analysis. In the Raman spectra, the color-filled peaks at 180 cm⁻¹ and 265 cm⁻¹ are assigned to $Cu_2SnSe_3$ and $Cu_xSe$, respectively[42,43]. **g**, **h** X-ray photoelectron spectra (XPS) spectra of SG and SLSG samples. Circles show the raw data, and lines give the muti-peak fitting results. Orange lines represent $Sn^{4+}$ and green lines represent $Sn^{2+}$. **i** Energy dispersive X-Ray fluorescence (XRF-EDX) elemental composition analysis of SG and SLSG samples. Error bar: standard deviation of the measurement results of a batch of samples.

charge transport ability, which could be attributed to excessive residual organics within the film. To address this issue, we further optimized the SLSG route by controlling the rate at which the Se concentration declined in the ripening stage. We refer to this optimized SLSG process as SLSG-O (Supplementary Fig. 12) and the detailed optimization process is shown in Supplementary Fig. 13.

Cross-sectional morphology and spectroscopy characterization of residual organics were performed on the SLSG and SLSG-O samples. As shown in Fig. 3b, the SLSG sample exhibited a double-layer structure consisting of a top layer with large grains and a bottom amorphous layer. Fourier transform infrared spectroscopy (FTIR) results (Fig. 3c) indicated that the amorphous layer in the SLSG sample was composed of a C-N framework, which exhibited a broad IR peak in the range of

1100–1300 cm⁻¹[49]. Additionally, no graphite signatures were observed in the Raman spectra (Supplementary Fig. 14). Therefore, this C-N framework is insulating and significantly influences the charge transport ability, unlike the graphite-like carbon framework reported previously[50]. In the SLSG-O sample, the FTIR spectra showed the disappearance of C-N bonding signals, indicating the effective removal of insulating organic residues. This removal of the C-N framework also facilitated the fusion of crystals in the bottom layer into larger grains (Fig. 3b).

Kelvin probe force microscopy (KPFM) was employed to study the contact potential difference (CPD) of the SLSG and SLSG-O samples (Fig. 3d, e). The results demonstrated a significant improvement in the uniformity of CPD, with the average CPD of the ACZTSSe film decreasing from 460 to 100 mV in the SLSG-O sample. This indicates an

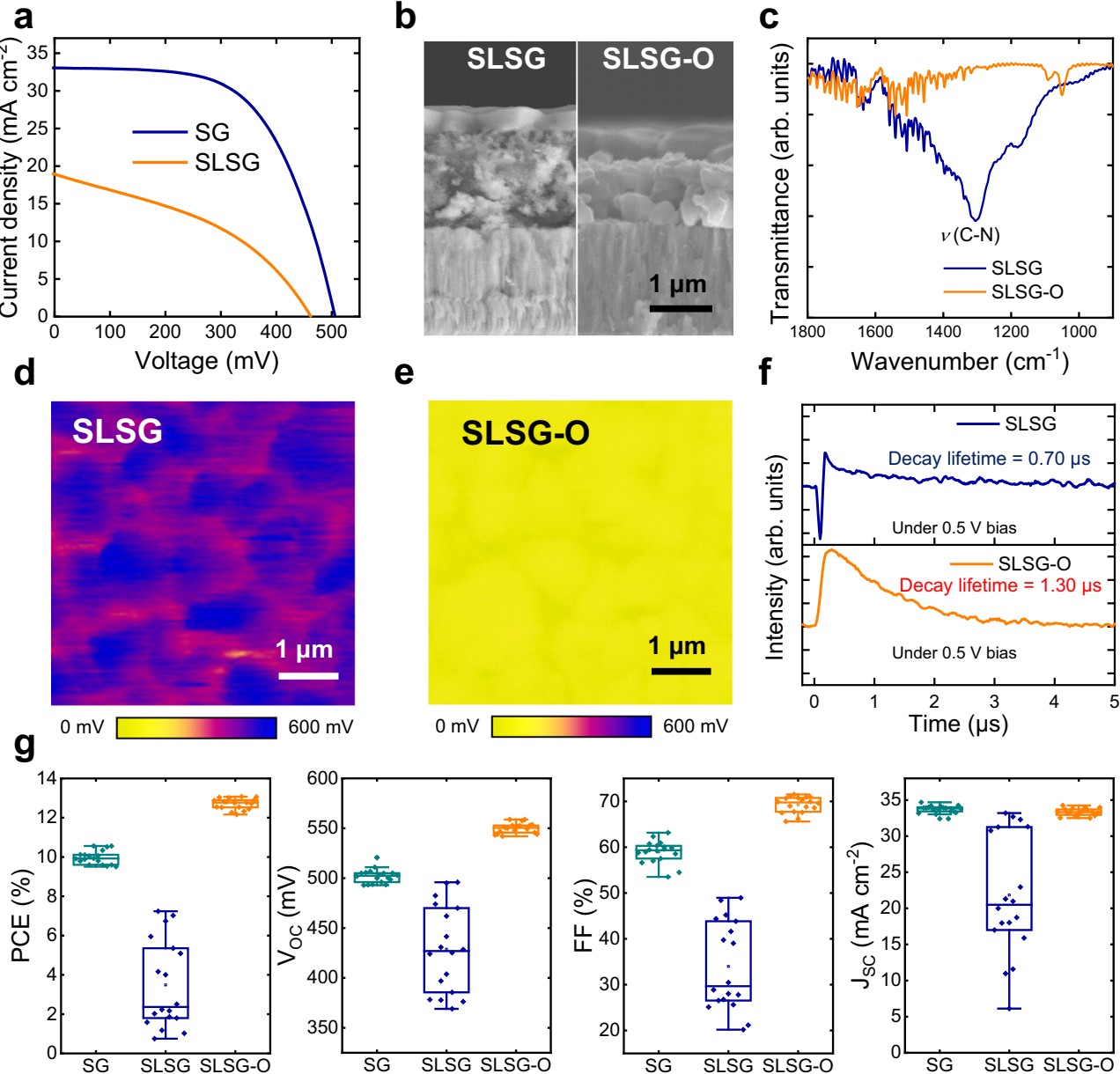

**Fig. 3 | Optimization of the SLSG route. a** Current density-voltage curves of the cells based on SG and SLSG routes. **b** Cross-sectional SEM images of SLSG and SLSG-O selenized films. **c** FTIR spectra of the SLSG and SLSG-O samples. The broad peak ranging from 1100 to 1300 cm⁻¹ in the SLSG sample belongs to C−N bonding, suggesting the existence of organic residues. The surface contacting potential difference (CPD) mappings of **d** SLSG and **e** SLSG-O films. The CPD of the SLSG-O sample is about 350 mV lower than that of the SLSG sample. **f** Transient photo-current curves of the SLSG and SLSG-O cells under 0.5 V. **g** Statistic analysis of device performance of SG, SLSG and SLSG-O solar cells.

enhancement in the p-type doping of the absorber film. Furthermore, the large negative signal observed in the transient photocurrent measured at 0.5 V (Fig. 3f) disappeared, suggesting the successful removal of the barrier to charge transport in the SLSG-O sample[51]. Finally, the average PCE of the SLSG-O devices exhibited a significant improvement (Fig. 3g & Supplementary Table 2), increasing from 10% to 12.6%, with the highest PCE reaching 13.1%. The highest open-circuit voltage ($V_{OC}$) reached 558 mV, attributed to the improved crystal quality. The average FF improved to 0.68, benefiting from enhanced charge transportability resulting from the effective removal of insulating organics.

### Defect properties and device performance
We conducted modulated electrical transient measurements to further investigate the influence of selenization routes on the performance of the final cells. This method can probe the charge transport and

recombination dynamics behaviors of a completed cell under different operational voltages and these charge dynamics properties can further be used to quantify the bulk and interface charge loss in the cell. The detailed experimental setup and analysis model have been described in our previous works[51,52]. The modulated transient photocurrent (M-TPC) measured at −1 V (Fig. 4a, b) displayed that the photocurrent decay of the SLSG-O sample was significantly faster compared to the SG sample. Additionally, the time position of the photocurrent peak for the SLSG-O sample remained almost independent of the applied voltage, whereas that of the SG sample was obviously shifted with increasing voltages. This indicates that the SLSG-O sample possesses a more stable and voltage-independent charge transport ability, primarily benefiting from improved crystallization quality, reduced presence of secondary phases and diminished carrier trapping states[52]. Distinct differences in carrier recombination properties between these

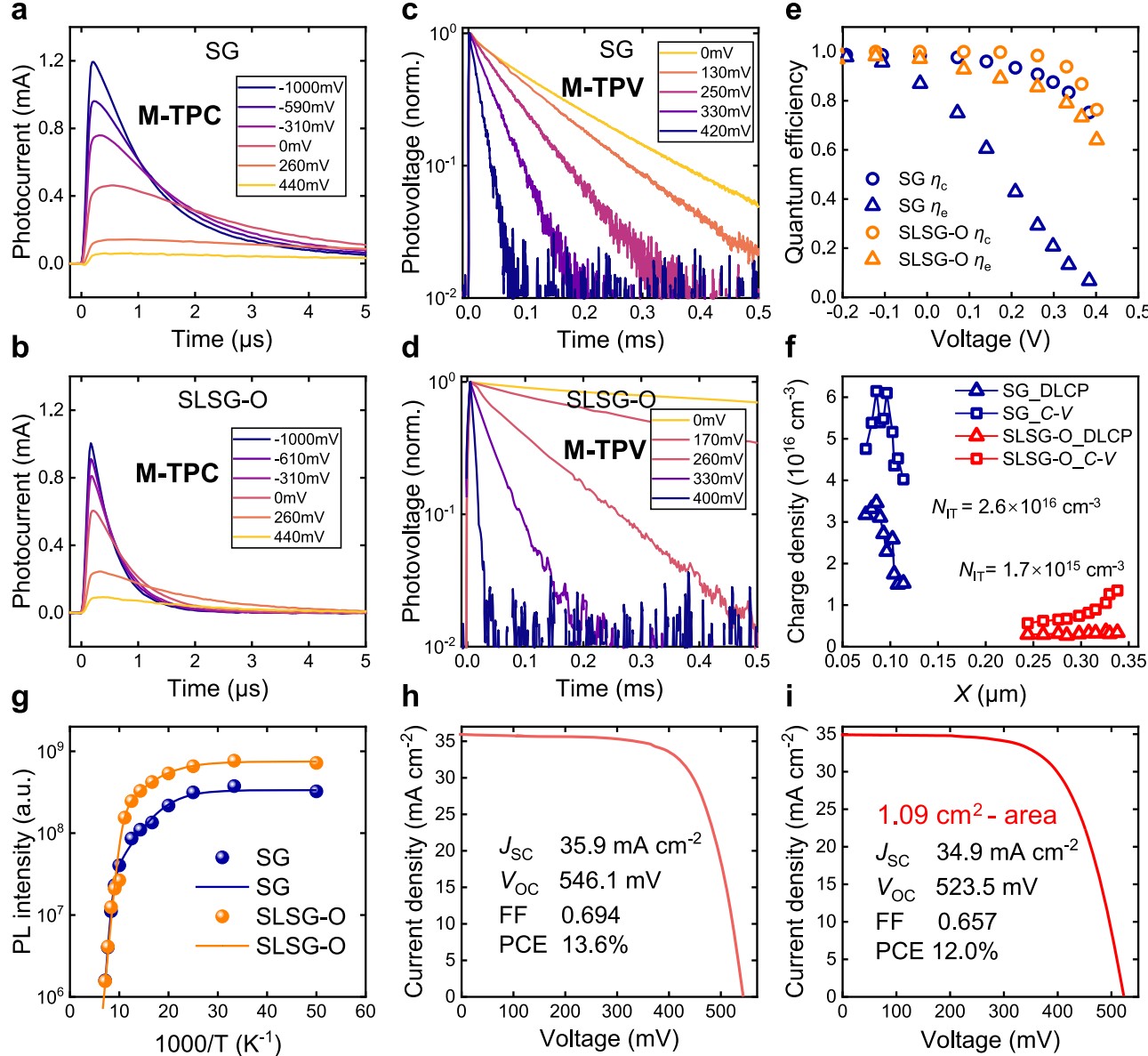

**Fig. 4 | Photoelectric characterization of final solar cells. a–d** Modulated transient photocurrent and photovoltage spectra of SG and SLSG-O solar cells. **e** Charge collection efficiency $\eta_c$ and extraction efficiency $\eta_e$ of the cells derived from the modulated electrical transient measurements. **f** *C-V* and DLCP characterizations of the SG and SLSG-O solar cells. The interface defect density ($N_{IT}$) is evaluated from the difference between the *C-V* and DLCP results. **g** Temperature-dependent integral PL intensity of the SG and SLSG-O films. Dots are the raw data and lines are fitting curves. SLSG-O film has higher PL intensity and larger PL quenching activation energy. Current density-voltage curves of the **h** small-area and **i** large-area (1.09 cm²) solar cells.

two samples were also observed in the modulated transient photovoltage (M-TPV) results in Fig. 4c, d. For the SG sample, the photovoltage measured at 0 V exhibited dual-exponential decay dynamics, with a fast decay in the early stage. This fast decay is typically caused by carrier recombination at the surface of the absorber film and at the heterojunction interface, possibly induced by defects related to Sn and Se. In contrast, the photovoltage decay of the SLSG-O sample exhibited a single exponential dynamic behavior with significantly longer lifetimes, indicating remarkable suppression of carrier recombination in the cell. Furthermore, the photovoltage lifetime of the SLSG-O sample displayed a more pronounced dependence on voltage, resulting in a smaller ideal factor compared to the SG sample (Supplementary Fig. 15). These results indicate that the SLSG-O route has improved the heterojunction properties of the cell by enhancing the p-type nature of the absorber and by improving the surface quality. Quantifying the modulated electrical transient measurements demonstrated a higher charge collection ($\eta_C$) efficiency in the SLSG-O sample (Fig. 4e)[53], confirming that the improvement in the heterojunction properties has reduced interface charge loss in the cell.

This result was further supported by other electrical characterizations (Fig. 4f & Supplementary Fig. 16). The interface defect density in the SG sample, estimated from the difference between drive-level charge profiling (DLCP) and capacitance-voltage profiling (C-V) results, was more than 15 times higher than that of the SLSG-O sample[54]. The charge profiling results further indicated a reduction in bulk defect density from $10^{16}$ to $10^{15}$ cm$^{-3}$. This reduction in bulk defect density agrees with the improved charge transport observed from the M-TPC, which resulted in a significant improvement in the charge extraction efficiency ($\eta_e$) of the SLSG-O cell, particularly in the high-voltage regime (Fig. 4e). The enhancement in the $\eta_e$ indicated that charge loss in the bulk absorber has been reduced, which primarily benefited from both the lower defect density and lower charge capturing velocity due

**Table 1 | Device performance parameters of efficient Kesterite solar cells**

| Device | PCE (%) | $J_{SC}$ (mA·cm$^{-2}$) | $V_{OC}$ (mV) | FF | $E_g$ (eV) | $E_g/q$-$V_{OC}$ (V) |
|---|---|---|---|---|---|---|
| IBM cell[14] | 12.6 | 35.2 | 513 | 0.698 | 1.13 | 0.617 |
| DGIST cell[32] | 12.6 | 35.4 | 541 | 0.659 | 1.13 | 0.589 |
| NJUPT cell[15] | 13.0 | 33.6 | 529 | 0.729 | 1.11 | 0.581 |
| This work | 13.6 | 35.9 | 546 | 0.694 | 1.10 | 0.554 |

to improved ACZTSSe lattice ordering. The reduction in carrier non-radiative recombination was also confirmed by temperature-dependent photoluminescence (PL) measurement, which exhibited significantly higher emission intensity (Fig. 4g) and an obvious spectral blue shift of approximately 40 meV (Supplementary Fig. 17)[55].

Finally, we achieved a high-performance solar cell with a total-area efficiency of 13.6% (certified at 13.44% in the National PV Industry Metrology and Testing Center (NPVM), as shown in Supplementary Fig. 18), which is among the highest reported results to date. The cross-sectional morphology of the completed cells with different ACZTSSe absorbers is shown in Supplementary Fig. 19. The current density-voltage (J-V) characteristics of the cell are depicted in Fig. 4h, and the external quantum efficiency spectrum (EQE) is provided in Supplementary Fig. 20. The bandgap ($E_g$) of the cell was determined to be 1.10 eV, and the integrated short-circuit current density ($J_{SC}$) is 37.0 mA·cm$^{-2}$, which closely aligns with the J-V result. The detailed performance parameters of our cell, as well as those of state-of-the-art Kesterite solar cells, are summarized in Table 1. Our cell exhibits a $V_{OC}$ of 546 mV, and the corresponding $V_{OC}$ deficit ($E_g/q$-$V_{OC}$) is 0.554 V. Notably, this $V_{OC}$ deficit value is significantly lower than that of the 12.6% and 13.0% record cells[14,15]. Moreover, our strategy also demonstrates high performance in large-area (1.09 cm$^2$) devices, achieving a champion PCE of 12.0% (certified at 12.1% in NPVM), as in Fig. 4i & Supplementary Fig. 21.

## Discussion

In this study, we have implemented a dual-temperature zone selenization route to realize a solid-liquid/solid-gas synergistic selenization reaction strategy. The introduction of a large amount of liquid Se has facilitated a solid-liquid reaction pathway, while the high Se chemical potential has promoted the direct and rapid formation of the Kesterite phase. In the subsequent stage, a synergistic regulation of Se condensation and volatilization has led to improved crystal growth and enhanced removal of organic residues. As a result, we have successfully achieved Kesterite films with reduced bulk and interface defects, leading to a remarkable device PCE of 13.6% and a large-area device PCE of 12.0%. Overall, our work represents a positive endeavor to precisely control the selenization reaction process and the multiphase reaction pathways. Decoupling the gas-phase reactant supply from the gas-solid reaction space also provides a wider window for diversifying control over reaction environments. In the future development of Kesterite solar cells, more types of strategies are desired to control solid-gas/solid-liquid reactions of this complex inorganic compound. In addition, achieving more accurate control over the reaction microenvironment, particularly in the alkaline metal environment, will allow for attaining higher device performance.

## Methods

### Reagents and materials

Thiourea (99%, Alfa), 2-Methoxyethanol (EGME) (99.8%, Aladdin), AgCl (99.5%, Innochem), CuCl (99.99%, Alfa), SnCl$_4$ (99.998%, Macklin), Zn(Ac)$_2$ (99.99%, Aladdin).

### ACZTSSe precursor film preparation

Firstly, 7.311 g thiourea was added into Vial 1 containing 15 ml EGME and stirred until dissolved. Then, 0.345 g AgCl, 2.16 g CuCl were successively added into Vial 1, and stirred till completely dissolved. Secondly, 15 ml EGME was injected into Vial 2 containing 3.963 g SnCl$_4$ under stirring. Thirdly, 3.126 g Zn(Ac)$_2$ was added into the SnCl$_4$-EGME suspension till thoroughly dissolved. Fourthly, mixed the solution in Vial 2 and Vial 1, and then obtained clear precursor solution. All the above were performed in glove box.

The filtered precursor solution was spin-coated onto a pre-cleaned Mo substrate at 2000 rpm for 30 s, followed by annealing on a hot plate at 280 °C for 2 min. This coating-annealing process was repeated 4 times in ambient conditions (25 °C, uncontrolled humidity, air). The Mo substrate was pre-cleaned using dish soap and deionized water (including 1 min ultrasonic cleaning in water) and was dried by N$_2$ gas flow. Then, in a dual-temperature zone tube furnace, two quartz boat filled with selenium particles (total weight is 26 g) was placed parallelly into Zone 1. and precursor films held by a thin graphite plate was placed into Zone 2, as in Fig. 1b. The whole selenization was performed under an inert atmosphere with constant N$_2$ flow of 300 sccm and under ambient pressure. The schematic time-dependent temperature curves of the Se source and CZTS precursor are shown in Fig. 1c. After the selenization, the whole system is naturally cooled to room temperature with ~150 min.

### Device fabrication

A 40-50 nm thickness CdS buffer layer was deposited on the top of ACZTSSe films by the chemical bath deposition (CBD) method, followed by sputtering 50 nm i-ZnO layer (sputtering power: 55 W, gas source: Ar, pressure: ~2 Pa, temperature: 60 °C) and 200 nm ITO layer (sputtering power: 60 W, gas source: Ar, pressure: ~0.2 Pa, temperature: 160 °C).

50 nm nickel (Ni) and 2 μm aluminum (Al) were evaporated on ITO layer. Finally, 110 nm MgF$_2$ layer covered the whole device, serving as the anti-reflection coating (ARC). The cells were separated from each other by mechanical scribing.

### Film Characterization

Raman spectra were carried out on Raman spectrometer (Lab-RAM HR Evolution, HORIBA) by using 532 nm laser diode as the excitation source. X-ray diffraction (XRD) patterns were collected by using an X-ray diffractometer with Cu Ka as the radiation source (Empyrean, PANaltcal). FTIR characterizations were performed by a Fourier Transform Infrared (FTIR) Spectrophotometer (TENSOR27, Bruker). Scanning electron microscopy (SEM) images were measured on Hitachi S4800 SEM using 10 kV power. Elemental ratios were determined by an energy dispersive X-Ray fluorescence (XRF-EDX) spectrometer (EDX-7000, Shimadzu). Kelvin probe force microscope (KPFM) images were obtained on an atomic force microscope (Multimode 9, Bruker). The XPS characterizations were performed on X-ray photoelectron spectrometer (Thermo Fisher Scientific ESCALAB 250Xi). Optical images of samples were measured on metallographic microscope (OLYMPUS, BX61). Steady-state photoluminescence (PL) spectra were obtained from PL spectrometer (Edinburgh Instruments, FLS 920), excited with a picosecond pulsed diode laser (EPL-640) with the wavelength of 638.2 nm while cooling down with liquid helium.

### Device Characterization

Modulated transient photocurrent and photovoltage (M-TPC/TPV) measurements were obtained by our lab-made setup[51], in which the cell

was excited by a tunable nanosecond laser pumped at 532 nm and recorded by a sub-nanosecond resolved digital oscilloscope (Tektronix, DPO 7104) with a sampling resistance of 50 Ω or 1 MΩ. A signal generator connected with a low-pass filter is parallel connected to the cell to give steady-state bias voltage at the cell. The current density-voltage curves were recorded on Keithley 2400 Source Meter under simulated AM 1.5 sunlight (100 mW·cm$^{-2}$) calibrated with a Si reference cell (calibrated by NIM). The voltage was forward scanned from −50 mV to 600 mV with a scanning rate of 90 mV·s$^{-1}$. The $J$-$V$ tests were conducted in air at 25 °C, and no preconditioning of the device was before the measurement. In the certification in NPVM, the aperture area of the small-area cell is 0.2627 cm$^2$ and the designated illuminated area of the large-area cell is 1.066 cm$^2$. For each statistic box in Fig. 3, 18 cells were involved. External quantum efficiency (EQE) was measured by Enlitech QE-R test system using calibrated Si and Ge diodes as references. The drive-level capacitance profiling (DLCP) was measured on an electrochemical workstation (Versa STAT3, Princeton) by using 11 kHz and 100 kHz AC excitation with amplitude from 10 to 100 mV and with DC bias from 0 to −0.5 V.

### Reporting summary

Further information on research design is available in the Nature Portfolio Reporting Summary linked to this article.

### Data availability

The main data supporting the findings of this study are available within the main text, Supplementary materials and source data files. Source data are provided with this paper.

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

## Acknowledgements

This work was financially supported by the National Natural Science Foundation of China (Grant nos. U2002216 (Q. M.), 52222212 (J. S.), 52172261 (Y. L.), 51972332 (H. W.), 52227803 (Q. M.)). J. S. also gratefully acknowledges the support from the Youth Innovation Promotion Association of the Chinese Academy of Sciences.

## Author contributions

X. X., J. S., Y. L., and Q. M. conceived the study. X. X. and J. Z. designed and conducted most of the experiments. X. X. and J. Z. contributed equally to this work. K. Y., J. W., L. L., M. J., B. Z., and D. L. participated in the data analysis and discussion. J. S. contributed to the experiment design and M-TPC/TPV data analysis. H. W. helped the Se vapor concentration and EQE measurements. J. S., Y. L., and Q. M. guided the study and supervised the execution. The manuscript is prepared, revised, and finalized by X. X., J. S., Y. L., and Q. M. All authors discussed the results and approved the manuscript.

## Competing interests

The authors declare no competing interests.
