## [Peer Review File · Nature Communications]

Controlling Selenization Equilibrium Enables High-Quality Kesterite Absorbers for Efficient Solar CellsREVIEWER COMMENTS

Reviewer #1 (Remarks to the Author):

This manuscript reports that the supply of initiate concentration of Se source during annealing process is importance to make a good quality of CZTSSe thin film. To control the amount of Se source precisely, they introduce the two-zone annealing system instead of single-zone system and effectively control a supplying amount of Se source on a precursor during the annealing process. This new system drives mitigating the defect formation in CZTSSe absorber and removing the insulative organic residue.

These results can be used to anneal other materials which needs an initiate Se amount or which needs to remove the organic residue on a surface.

Although its necessity of this new system and these results, here several argues can be raised, especially for the reaction mechanism part.

1. What about the geometric construction of Se concentration simulation for two-zone system?

Figure S1(a) shows the simulation result of single-zone graphite box system.

What about the two-zone system? Because Se is carried by N₂ gas from zone 1 to zone 2, people can easily think that the amount of Se at the position of B and C in a precursor sample can also be different: the sample area closed to the zone 1 (point B) could have more Se than the sample area distant from zone 1 (point C).

Did you also perform the simulation with two-zone system?

2. Missing the explanation for Figure S2 in the manuscript.

3. About the analysis of Raman and XRD data.

3-1) "Specifically, the Cu_xSe binary phase is firstly formed at 400 C"

- According to Figure 2(b1), you already marked CZTSSe and CZTS together with Cu_xSe. It means that CZTSSe is already formed in the SG_400C sample, passing through the reaction process of Cu_xSSe + ZnSSe + SnSSe(or SnSSe₂) -> CTSSe + ZnSSe -> CZTSSe. It is difficult to say that Cu_xSe is "FIRSTLY" formed at 400C.

- Additionally, according to the XRD data described in Figure S5(a), it does not show any kinds of CZTSSe, CZTS, Se or any other peaks. Yes, I understand that those phases would be existed in the sample as amorphous phases. Therefore, here I suggest measuring the cross-sectional EDS mapping with this sample. Then, it will be clear that the film consists of CZTSSe phase with the Cu_xSe phase in the SG_400C sample.

3-2) Please enlarge the XRD data in Figure S5. Mo peak at ~40.5 deg. is not the important information in this data.

- You measure XRD data to confirm whether the secondary phases are existed in the samples or not. However, with this magnification of XRD data in Figure S5, it is difficult to see whether there are small peaks in there or not. Please refer ACS Omega 2020, 5, 10501, Figure 5 ("Characterization of Cu₂ZnSnS₄ particles obtained ...") or ACS Appl. Mater. Interfaces 2021, 13, 13425, Figure 2 ("Influence of the reaction pathway on the defect formation ...").

3-3) Existence of Se phase in the sample SLSG_540C (Figure 2(a2))

- As you mentioned, Figure 2(a2) is SEM image of SLSG sample taken at 540 C for 200 s and mentioned that the area with blue color is the liquid Se. However, Se cannot be existed as a single Se phase at 540 C because its melting point is ~220 C and because it can easily react with other cations at this temperature. Additionally, I cannot find any Se peaks in Figure S5(b) for SLSG-500C data (blue color) although the Se peaks are clearly observed in Figure S5(a) for SLSG-400C data (blue color). (Se peaks are nearby ~23 and ~30 deg. Therefore, if you would like to assert that the blue area in Figure 2(a2) is the liquid Se phase, I recommend you measure AES spectroscopy with this sample.

3-4) For the same reason with 3-3), Raman data in Figure 2(c2) do not represent the Se phase at ~240 and ~250 cm⁻¹.

- As you can see in J. Phys. Energy 2 (2020) 012002, Figure 7(c) with green color (“Point defects, compositional fluctuations, and secondary phases ...”), it is general CZTSSe (Se-rich) Raman peaks. Additionally, XRD data in Figure S5(b) do not show any diffractions related with Se phase. Please check the Raman data more in details.

3-5) Please enlarge the XRD data in Figure S5. Mo peak at ~40.5 deg. is not the important information in this data.

- You measure XRD data to confirm whether the secondary phases are existed in the samples or not. However, with this magnification of XRD data, it is difficult to distinguish whether there are small amounts of other phases or not. Please refer ACS Omega 2020, 5, 10501, Figure 5 (“Characterization of Cu₂ZnSnS₄ particles obtained ...”) or ACS Appl. Mater. Interfaces 2021, 13, 13425, Figure 2 (“Influence of the reaction pathway on the defect formation ...”).

3-6) Existence of CTSe in SLSB-500C (Figure S4(b)).

- Referring to your previous paper (Nature Energy, 2023, <https://doi.org/10.1038/s41560-023-01251-6>, “Control of the phase evolution of kesterite ...”) in Figure 1(b), the Raman data with ACZTSSe at 535 C, this peak is similar with the peak described in Figure S4(b) of this current manuscript. Although those peaks are similar, especially 170–190 cm⁻¹ regions, you said that there is no CTSe in Nature Energy, but now you said that there is a CTSe phase in this manuscript. Therefore, it needs the explanation.

4. About the appearance of decomposition on a surface of CZTSSe film.

- I totally agree that the SLSG route is able to suppress the appearance of the Sn vacancy deep defect, but I cannot agree that the decomposition appears on a surface of the CZTSSe film because of following reasons.

reason 1) Sn^{2+} can also be appears in CZTSSe phase when the CZTSSe is formed from SnSe instead of SnSe_2 .

reason 2) XRF-EDS data (Figure 2(e)) represents not only the surface of the CZTSSe film, but also in-depth of the CZTSSe film. That means, the decrease of Sn/Zn and Sn/(Cu+Zn+Sn) ratio of SG sample in comparison with SLSG sample do not represent the decomposition of CZTSSe phase on a surface of the film. It just shows that the Sn concentration of SG sample is less than that of SLSG sample.

reason 3) Consequently, I can induce that the SLSG system can get a sufficient Se amount during the annealing process, so that the SnSe_2 phase can be easily formed in comparison with the SG system. (According to the Sn-Se phase diagram, SnSe_2 can be formed when the Se at% increase, while SnSe is formed when the Se at% decrease.) Therefore the CZTSSe phase of your SLSG sample can have Sn^{4+} cation more than that of SG sample, resulting mitigation of Sn defect formation in the film.

Please consider these points before you publish your valuable data.

Reviewer #2 (Remarks to the Author):

See attachment.

Dear Editor,

This paper reports 13.6 % CZTSSe solar cells with anti-reflection coating. The 13.6% is really impressive value to the research community of kesterite solar cell. However, major revisions are necessary for this paper to be accepted.

1. Up to now several papers reported ~13% PCE of kesterite solar cell using a graphite box using one-zone furnace:
 - 1) Son, D. H. et al. Effect of solid-H₂S gas reactions on CZTSSe thin film growth and photovoltaic properties of a 12.62% efficiency device. *J. Mater. Chem.* **7**, 25279–25289 (2019)
 - 2) Li, J., Huang, J., Ma, F. *et al.* Unveiling microscopic carrier loss mechanisms in 12% efficient Cu₂ZnSnSe₄ solar cells. *Nat Energy* **7**, 754–764 (2022)
 - 3) J. Zhou et. al, “Control of the phase evolution of kesterite by tuning of the selenium partial pressure for solar cells with 13.8% certified efficiency” *Nature Energy*. (2023) <https://doi.org/10.1038/s41560-023-01251-6>
 - 4) Y. Gong et al., Elemental de-mixing-induced epitaxial kesterite/CdS interface enabling 13%-efficiency kesterite solar cells. *Nat. Energy* **7**, 966–977 (2022).
 - 5) K. Yin et al., Lanthanum-induced synergetic carrier doping of heterojunction to achieve high-efficiency kesterite solar cells, *J. Mater. Chem. A*, (2023) <https://doi.org/10.1039/D3TA00597F>

In above papers, last three papers are reported by Prof. Qingbo Meng group, who is the corresponding author of this manuscript. ~13% PCE is also achieved by using a graphite box using one-zone furnace.

In this manuscript, authors also reported over 13% using two-zone furnace. Then, the strength of using two-zone furnace should be more stressed. Authors only showed statistical data of devices in Fig. 3 (f1-f4). How many devices are analyzed in the Fig. 3(f1-f4)? Can you provide a table of device parameters for Fig. 3(f1-f4)?

Furthermore, in the temperature profile in Fig. S8, for “SLSG-O” process, authors should show some acceptable window range to achieve high PCE. Authors used 350 °C for S_{ex} condensation and t₀= 300s for optimal SLSG absorber. How much are they sensitive for a higher PCE cell?

Authors add some acceptable temperature range for the 350 °C and to= 300s. Another important one is that you added “SLSG-O” as an optimized declining rate. You should add some window for the acceptable temperature declining rate as in the following figure with supporting information of measured device data.

If the window is very narrow, or device performance is very sensitive to the temperature profiles, people may not use your two zone furnace method. Because people including your group already produce 13 % PCE solar cell using a graphite box using one zone-furnace. Small window may reduce novelty of your method, and this is the most critical point in revisions.

2. In addition to the review comment #1, for the thermal cycle of selenization process, duration time at highest temperature and cooling temperature profiles for Se-source zone and CZTS precursor zone (after ~ 3500 s) should be also clearly displayed and discussed.
3. In Figure 1b, schematic of quartz furnace is shown. Can you add diameter of quartz diameter, the distance between Se-source (in Zone 1) and precursor film (in Zone 2)? Further, can you provide the distance between gas inlet and Se-source and the distance between precursor film and gas outlet. You also add some discussions how much margins are allowed in the distance for the high PCE? Much are not clear about your SLSG process.
4. In the two zone furnace, 26 g of Se is placed both at Zone-1 and Zone-2. The Se amount affects the Se vapor pressure near precursor film. Can you also add acceptable Se-amount for the higher PCE? Se-partial pressure is important to obtain high PCE cell. As in your paper (Control of the phase evolution of kesterite by tuning of the selenium partial pressure for solar cells with 13.8% certified efficiency, Nature Energy), over- pressure (such as 1.6 atm) enables PCE more than 13 %. Can you add more information about the Se-vapor pressure too?
5. In Figure 2, Raman data Cu_xSe and Cu_2SnSe_3 were used for fitting, In Figure S4, Se was

used for Raman fitting. Can you provide XRD data for supporting the presence of those secondary phases? In the XRD data of Fig. S5, peaks are not assigned.

6. Generally, in chalcogenide, Cu_xSe is regarded as a flux to increase the grain size. You addressed the liquid Se assisted selenization of CZTSSe (as in the Figure S7). Can you discuss more about the difference between liquid Se-assisted growth and liquid Cu_xSe -assisted growth?
7. In this manuscript, some FE-SEM images are presented. However, no EDS data are shown. Raman and XRD are presented for the phase evolution. Along with the data of the structural analysis, chemical compositions should be provided with EDS result. EDS result provide information of how much Se covers the films and how much Sn loss occurs at each selenization stages.
8. Authors claimed 13.6 % for small cell and 12 % for large cell (over 1cm^2). However, in the certificated record in the supporting information, only PCE of small cell was displayed. Best PCE of small cell was shown as 13.45 % not 13.6 % in the certification. 13.45 % is better to be claimed. For the large cell PCE, it should be also presented by official certification with exact cell area and a photo of measured device.

Reviewer #3 (Remarks to the Author):

The manuscript by Xu et al is interesting as it beats the previous efficiency record of 13% by providing a cell with an efficiency of 13.4% for CZTSSe. This is impressive and justifies the publication of the manuscript. However, I do have severe concerns as the manuscript does not provide adequate details of their experiments. Therefore, I would like the authors to answer the following questions:

1. in the Methods section, it is stated that AgCl was added along with the other reactants. If that is true, then they do not have CZTSSe but $(\text{Ag, Cu})_2\text{ZnSn}(\text{S,Se})_4$ i.e. ACZTSSe. If this is correct then the authors should not only change the title but state it upfront in the manuscript. Furthermore, there are some early papers published on alloying with Ag that show improvement in carrier lifetime etc. and authors should cite pertinent papers from the literature. The efficiency results even after alloying with silver are still interesting but warrant a different discussion. (For example, rather than attributing the improvement entirely to selenization condition, it will be fair to say that improvement may be in addition due to alloying with silver).

2. The details of the calculation profile in Figure S1 are needed. What are the pertinent equations? The details in Table S1 are not sufficient. It is not clear why the diffusion coefficient of water vapor in air and water density are listed in Table S1. By the same token, the diffusion coefficient for selenium is missing that must have been used in the simulation. Points A, B, C in Figure S1 really do not have sensors, do they? Currently, it implies that Se concentration at those points was measured. Furthermore, point A, must have been assumed to be in equilibrium with the Se droplet, if yes, then why is Se concentration rising with time at point A (is this due to changing temperature of the Se droplet)? All this confusion arises because boundary conditions in the simulation are not explicitly provided along with the equations.

3. We need more information regarding their equipment setup. How long is their furnace tube? What was the distance between Zone 1 and Zone 2 in Figure 1 (b)? Is the precursor film along with its substrate put on top of another glass (the blue item shown in the Figure)?

4. t_p in Figure 1 (c) should be t_0 .

5. The authors talk about all the grain growth etc. in detail with reference to SG and SLSG. However, SLSG gave poor cell performance and they shifted to SLSG-O. However, very little detail is provided for SLSG-O. Are they referring to decreasing the temperature of the Se source around 2000 S (I am assuming it to be in seconds – they should list proper units for both the X and Y axis in this figure) as shown in Figure 1(c)? If yes, then is the figure drawn to scale? Specifically, they should tell us how fast the Se source temperature decreased (the initial and final temperatures with the time duration).

6. An important item missing is the final cooling step. After the completion of selenization at 540C, how were the sample and the Se source cooled? This profile is missing in the text and is very much needed.

7. They should tell us how the blue region in Figure 2(a2) was deduced to be cooled liquid Se.

8. In Figure 3f, how many solar cells were studied? The deviation is over how many samples? Do the samples include cells from different batches or are they from the same overall film (i.e. six cells from their one coated film shown in S1)? This is important to understand reproducibility.

9. It seems that their reported efficiencies are based on the total cell area. It will be good to state that once in the main text.

10. On page 13, they state 'while the rise of TPC signal of the SLSG-O sample is obviously slowed down ...'. It is not obvious at all. They need to explain to clarify for the reader what they mean by it.

11. For the large area device, they need to show its picture. How are the grids patterned? It will help in assessing that indeed current was collected from the large area.

Overall, I do believe the results are interesting and warrant publication. However, for the reader to understand their improvement and replicate their experiment, it is essential that they do provide important details. They need to make major revisions.

Reply to Reviewer #1

Firstly, we would like to show our sincere appreciation to your kind reviewing, professional comments and suggestions. As you suggested, this manuscript has been thoroughly revised and detailed questions you proposed will be answered one by one in the following.

1. What about the geometric construction of Se concentration simulation for two-zone system? Figure S1(a) shows the simulation result of single-zone graphite box system. What about the two-zone system? Because Se is carried by N₂ gas from zone 1 to zone 2, people can easily think that the amount of Se at the position of B and C in a precursor sample can also be different: the sample area closed to the zone 1 (point B) could have more Se than the sample area distant from zone 1 (point C).

Did you also perform the simulation with two-zone system?

Reply: Thanks very much for your valuable question. We have made a Se concentration simulation of the two-zone system in the manuscript revision process in Figure S1. The geometric construction of this system is shown as below (Figure S2), and more detailed parameters used in the simulation is given in the supporting information.

Figure S2

Figure S1. (a, c) graphite box (scale: mm), (b, d) two-zone system (scale: cm)

The simulation results are presented here in the form of time-dependent Se concentration at representative points for clarity (A, D: close to the Se source, B, E: left edge of the sample, C, F: right edge of the sample). As seen in the above figure (right one), in the two-zone system, the Se concentration in the sample region exhibits a rapid increase in the initial 70 s and keeps constant in the following stage. Moreover, the Se exhibits a uniform distribution in the sample region and no obvious difference in the concentration between positions D and E can be observed from the simulation. This phenomenon is obviously different from single-zone graphite box system, which shows very slow increase in the Se concentration and non-uniformity Se distribution in the sample region. This result is reasonable because in two-zone system the gas carrier has facilitated the Se transport, which weakens the non-uniformity arisen from spontaneous diffusion.

2. Missing the explanation for Figure S2 in the manuscript.

Reply: Thanks very much for your kind remind. The order of the Figure S2 has been revised

to Figure S3 and we have added a description of this figure as “The temperature evolutions of these two selenization modes are schematically depicted in Figure S3.”

3. About the analysis of Raman and XRD data.

3-1) “Specifically, the Cu_xSe binary phase is firstly formed at 400 C”

- According to Figure 2(b1), you already marked CZTSSe and CZTS together with Cu_xSe . It means that CZTSSe is already formed in the SG_400C sample, passing through the reaction process of $\text{Cu}_x\text{SSe} + \text{ZnSSe} + \text{SnSSe}(\text{or SnSSe}_2) \rightarrow \text{CTSSe} + \text{ZnSSe} \rightarrow \text{CZTSSe}$. It is difficult to say that Cu_xSe is “FIRSTLY” formed at 400C.

- Additionally, according to the XRD data described in Figure S5(a), it does not show any kinds of CZTSSe, CZTS, Se or any other peaks. Yes, I understand that those phases would be existed in the sample as amorphous phases. Therefore, here I suggest measuring the cross-sectional EDS mapping with this sample. Then, it will be clear that the film consists of CZTSSe phase with the Cu_xSe phase in the SG_400C sample.

Reply: Thanks very much for your comment. Indeed, the statement “the Cu_xSe binary phase is firstly formed at 400 C” is not appropriate here. We have deleted this sentence and this deletion will not influence the description and understanding of the Raman results.

As you suggested, we have also measured EDX mapping of the SG-400 °C samples, and the results are shown as below. We should honestly say that no local accumulation of Cu and Se can be observed in the EDX mapping, and all the metal elements exhibit homogeneous distribution. This phenomenon is reasonable because the low temperature (400 °C) and short heating duration cannot drive the obvious metal atom/ion transport and local accumulation. Nonetheless, according to the Raman and XRD results (Figure S7 in the revised version and the next answer below), we are sure that the Cu_xSe phase has formed, which may exist in the form of nanocrystals and cannot be distinguished by SEM EDX.

3-2) Please enlarge the XRD data in Figure S5. Mo peak at ~ 40.5 deg. is not the important information in this data.

- You measure XRD data to confirm whether the secondary phases are existed in the samples or not. However, with this magnification of XRD data in Figure S5, it is difficult to see whether there are small peaks in there or not. Please refer ACS Omega 2020, 5, 10501, Figure 5 (“Characterization of $\text{Cu}_2\text{ZnSnS}_4$ particles obtained ...”) or ACS Appl. Mater. Interfaces 2021, 13, 13425, Figure 2 (“Influence of the reaction pathway on the defect formation ...”).

Reply: Thanks very much for this comment. We have amplified the patterns and the peaks corresponding to different phases have been marked in the revision for clarity, as shown below (Figure S7 in the revised version). These two references are very valuable for us to identify the XRD peaks and we have also cited them in the revision manuscript. A small peak corresponding to the Cu_xSe phase has indeed been observed in the SG sample, which is consistent with the Raman result.

Figure S7. XRD patterns of SG and SLSG samples in different intermediate stages: (a) 400 °C and (b) 500 °C. Three characteristics XRD peaks of Se_x can be clearly seen in the SLSG sample ($\sim 23.6^\circ$, 29.8° , 43.7° , referring to JADE PDF#06-0362). In the SG-400 °C sample, a small XRD peak corresponding to the Cu_{2-x}Se phase can be seen at about 44.3° (JADE PDF#06-0680), which is consistent with the Raman result. In addition, the SLSG sample shows more obvious CZTSSe XRD peaks, indicating that the crystallization of CZTSSe has been facilitated.

3-3) Existence of Se phase in the sample SLSG_540C (Figure 2(a2))

- As you mentioned, Figure 2(a2) is SEM image of SLSG sample taken at 540 C for 200 s and mentioned that the area with blue color is the liquid Se. However, Se cannot be existed as a single Se phase at 540 C because its melting point is ~ 220 C and because it can easily react with other cations at this temperature. Additionally, I cannot find any Se peaks in Figure S5(b) for SLSG-500C data (blue color) although the Se peaks are clearly observed in Figure S5(a) for SLSG-400C data (blue color). (Se peaks are nearby ~ 23 and ~ 30 deg. Therefore, if you would like to assert that the blue area in Figure 2(a2) is the liquid Se phase, I recommend you measure AES spectroscopy with this sample.

Reply: Thanks very much for this question. As you suggested, we have used SEM EDX and photograph of the practical film in the selenization process to demonstrate the existence of liquid Se on the film surface.

Figure: In-situ taking photos during the selenization process

Figure S6.

The photographs of the selenized samples in-situ taken during the selenization process (540 °C for 300 s) are shown in the above figure (Figure S6). It can be seen that when the sample adopts a SG selenization, its surface exhibits grey color due to the formed crystals. When the sample adopts the SLSG route, the sample surface initially exhibits black color, which is identical to that of the molten liquid Se. When the selenization continues, the black color region gradually shrinks, and the grey color is exposed in some regions. This is a typical liquid contraction behavior. Therefore, we can confirm that the liquid Se can retain on the sample surface even after reaching 540 °C. The liquid Se gradually volatilizes in this process. The continuously provided high-concentration Se in the selenization chamber has helped suppress the fast volatilization of the surface liquid Se and sustains the surface liquid Se for a relatively long time.

In this process, when we interrupted the high-temperature process and naturally cool the system to room temperature, surface Se can also be observed from SEM EDX. As shown in the below figure, in the SLSG sample, high Se EDX intensity can be seen in the left side of the film while all the metal elements show relatively lower EDX intensity in this region. In contrast, in the SG sample, the Se exhibits a synchronous distribution with the metal elements. These results indicate that elemental Se exists on the film surface. As such, we can confirm the liquid Se is retained in the selenization process. We have revised Figure 2(a-b) with these new SEM figures and the EDX mapping is given in Figure S5.

Figure S5

3-4) For the same reason with 3-3), Raman data in Figure 2(c2) do not represent the Se phase at ~240 and ~250 cm⁻¹.

- As you can see in J. Phys. Energy 2 (2020) 012002, Figure 7(c) with green color (“Point defects, compositional fluctuations, and secondary phases ...”), it is general CZTSSe (Se-rich) Raman peaks. Additionally, XRD data in Figure S5(b) do not show any diffractions related with Se phase. Please check the Raman data more in details.

Reply: Thanks very much for your question. Indeed, as you said, CZTSSe also has Raman peaks at ~240 and 250 cm⁻¹. However, in CZTSSe, these two peaks are usually much weaker than other peaks at ~173 and 198 cm⁻¹. This is obviously different from our measured Raman results. Thus, we prefer to assign the ~240 and 250 cm⁻¹ Raman peaks to the Se, which agrees with the reported Se Raman spectra (J. Appl. Phys. 120, 135101 (2016); New J. Chem., 2011, 35, 453–460).

As shown in the revised Figure S7(b), the XRD peaks corresponding to Se have also been marked for clarity.

3-5) Please enlarge the XRD data in Figure S5. Mo peak at ~40.5 deg. is not the important information in this data.

- You measure XRD data to confirm whether the secondary phases are existed in the samples or not. However, with this magnification of XRD data, it is difficult to distinguish whether there are small amounts of other phases or not. Please refer ACS Omega 2020, 5, 10501, Figure 5 (“Characterization of Cu₂ZnSnS₄ particles obtained ...”) or ACS Appl. Mater. Interfaces 2021, 13, 13425, Figure 2 (“Influence of the reaction pathway on the defect formation ...”).

Reply: Thanks very much for this question. The answer has been given in the 3-2).

3-6) Existence of CTSe in SLSB-500C (Figure S4(b)).

- Referring to your previous paper (Nature Energy, 2023, <https://doi.org/10.1038/s41560-023-01251-6>, “Control of the phase evolution of kesterite ...”) in Figure 1(b), the Raman data with ACZTSSe at 535 C, this peak is similar with the peak described in Figure S4(b) of this current manuscript. Although those peaks are similar, especially 170–190 cm⁻¹ regions, you said that there is no CTSe in Nature Energy, but now you said that there is a CTSe phase in this

manuscript. Therefore, it needs the explanation.

Reply: Thanks very much for this question and your careful reviewing. In fact, in the nature energy paper, the CTSe phase has always been considered in all the fittings in Figure 1(B). In the 535 sample, the CTSe Raman peak also exists with very weak intensity. For the Raman analysis, whether the CTSe phase should be considered or not mainly depends on the fitting quality and the spectra feature. The intensity (I) of the peak at 180 cm^{-1} relative to that at $\sim 173\text{ cm}^{-1}$ can be used as an indication to evaluate whether the CTSe phase is needed involved in the fitting. In the nature energy paper, $I(180\text{ cm}^{-1}) / I(173\text{ cm}^{-1}) = 0.77$, which is a little smaller than that ($I(180\text{ cm}^{-1}) / I(173\text{ cm}^{-1}) = 0.82$) in this manuscript (revised Figure S8(b)). For the SLSG-500 °C sample shown in Figure 2, $I(180\text{ cm}^{-1}) / I(173\text{ cm}^{-1}) = 0.68$. This obvious difference in the intensity ratio demands us to consider the CTSe phase in the Raman fitting in this work.

We also tried to fit the spectrum in the range between 150 to 210 cm^{-1} only by two CZTSSe peaks as in Figure (b), but cannot get satisfied fitting quality compared to that when the CTSe phase is also considered as in Figure (a). This is further demonstrated by the conventional residual results in Figure (c). Therefore, we think CTSe phase needs be considered here to get a more reliable analysis of the Raman spectra.

Figure: comparison of fitting result with or without peak at 180 cm^{-1}

4. About the appearance of decomposition on a surface of CZTSSe film.

- I totally agree that the SLSG route is able to suppress the appearance of the Sn vacancy deep defect, but I cannot agree that the decomposition appears on a surface of the CZTSSe film because of following reasons.

reason 1) Sn^{2+} can also be appears in CZTSSe phase when the CZTSSe is formed from SnSe

instead of SnSe₂.

reason 2) XRF-EDS data (Figure 2(e)) represents not only the surface of the CZTSSe film, but also in-depth of the CZTSSe film. That means, the decrease of Sn/Zn and Sn/(Cu+Zn+Sn) ratio of SG sample in comparison with SLSG sample do not represent the decomposition of CZTSSe phase on a surface of the film. It just shows that the Sn concentration of SG sample is less than that of SLSG sample.

reason 3) Consequently, I can induce that the SLSG system can get a sufficient Se amount during the annealing process, so that the SnSe₂ phase can be easily formed in comparison with the SG system. (According to the Sn-Se phase diagram, SnSe₂ can be formed when the Se at% increase, while SnSe is formed when the Se at% decrease.) Therefore the CZTSSe phase of your SLSG sample can have Sn⁴⁺ cation more than that of SG sample, resulting mitigation of Sn defect formation in the film.

Reply: Thanks very much for your comment and valuable ideas. We agree with you that the appearance of Sn²⁺ is not simply arisen from CZTSSe decomposition. We have revised this part in the manuscript, as shown below.

“The appearance of the Sn²⁺ cation could come from several factors. Firstly, the insufficient Se in the SG sample makes the CZTSSe surface in a Se deficiency condition, which leads more electrons to accumulate around the Sn, resulting in the Sn²⁺. Secondly, the insufficient Se makes some of the Sn-S precursor transform into SnSe, which also influences the Sn valence when the SnSe is reacted into CZTSSe. Thirdly, the Se deficiency in the CZTSSe decreases the coordination number of the Sn atom, making the CZTSSe easy to decompose at high temperatures, resulting in the SnSe decomposition product.^{5,27} Overall, the appearance of the Sn²⁺ cation is a direct result of the insufficient Se in the SG route.”

We hope these three factors can reasonably explain the appearance of the Sn²⁺.

Reply to Reviewer #2

Firstly, we would like to show our sincere appreciation to your kind reviewing, professional comments and suggestions. As you suggested, this manuscript has been thoroughly revised and detailed questions you proposed will be answered one by one in the following.

1. Up to now several papers reported ~13% PCE of kesterite solar cell using a graphite box using one-zone furnace:

1) Son, D. H. et al. Effect of solid-H₂S gas reactions on CZTSSe thin film growth and photovoltaic properties of a 12.62% efficiency device. *J. Mater. Chem.* 7, 25279–25289 (2019)

2) Li, J., Huang, J., Ma, F. et al. Unveiling microscopic carrier loss mechanisms in 12% efficient Cu₂ZnSnSe₄ solar cells. *Nat Energy* 7, 754–764 (2022)

3) J. Zhou et. al, “Control of the phase evolution of kesterite by tuning of the selenium partial pressure for solar cells with 13.8% certified efficiency” *Nature Energy.* (2023)
<https://doi.org/10.1038/s41560-023-01251-6>

4) Y. Gong et al., Elemental de-mixing-induced epitaxial kesterite/CdS interface enabling 13%-efficiency kesterite solar cells. *Nat. Energy* 7, 966–977 (2022).

5) K. Yin et al., Lanthanum-induced synergetic carrier doping of heterojunction to achieve high-efficiency kesterite solar cells, *J. Mater. Chem. A*, (2023)
<https://doi.org/10.1039/D3TA00597F>

In above papers, last three papers are reported by Prof. Qingbo Meng group, who is the corresponding author of this manuscript. ~13% PCE is also achieved by using a graphite box using one-zone furnace.

In this manuscript, authors also reported over 13% using two-zone furnace. Then, the strength of using two-zone furnace should be more stressed. Authors only showed statistical data of devices in Fig. 3 (f1-f4). How many devices are analyzed in the Fig. 3(f1-f4)? Can you provide a table of device parameters for Fig. 3(f1-f4)?

Reply: Thanks very much for this question. Indeed, several works have also reported the ~13% efficiency by graphite box. We also agree that in current stage, the graphite box may be enough to provide acceptable environment for selenization. Currently, the efficiency of CZTSSe solar

cells is currently limited by unidentified and uncontrollable defects, which results in no significant efficiency differences between varied selenization routes. However, the disadvantage of graphite box is obvious, which can hardly support the long-term development of CZTSSe solar cells, especially when we need much larger cell size and more controllable Se atmosphere. This is the aim of this manuscript, for more controllable selenization and the long-term development toward commercialization. Our approach proposed in this work is a preliminary effort, and more selenization strategies are all desired for this field.

Here we can make a detailed comparison with the last three papers that we have reported previously. Firstly, the selenization route in this work is performed in ambient atmosphere pressure, which have higher safety and operability than the high-pressure selenization that we have reported (3) and thus can be more easily expanded in the future works. Compared to the NJUPT cell (4), our cell has a lower voltage loss which may benefit from the more controllable selenization in this work. In our lab, if using the ambient-pressure graphite box selenization (5), the efficiency of standard cells is also <13%. Thus, we think the selenization route reported in this work has more potential to achieve more higher performance.

Regarding Figure 3, 18 cells have been involved in this statistic analysis. For clarity, the performance data have been added in the box figure, and a table of the device performance has also been added in the Table S2.

Furthermore, in the temperature profile in Fig. S8, for “SLSG-O” process, authors should show some acceptable window range to achieve high PCE. Authors used 350 oC for Sex condensation and to= 300s for optimal SLSG absorber. How much are they sensitive for a higher PCE cell? Authors add some acceptable temperature range for the 350 oC and to= 300s. Another important one is that you added “SLSG-O” as an optimized declining rate. You should

add some window for the acceptable temperature declining rate as in the following figure with supporting information of measured device data. (See attached document for a supplementary figure)

If the window is very narrow, or device performance is very sensitive to the temperature profiles, people may not use your two zone furnace method. Because people including your group already produce 13 % PCE solar cell using a graphite box using one zone-furnace. Small window may reduce novelty of your method, and this is the most critical point in revisions.

Reply: Thanks very much for this question. In fact, we had optimized some experimental conditions to obtain the champion cell. Nonetheless, the 350 °C parameter was used according to previous experience, and we did not optimize this parameter. At this temperature, the microstructure of the precursor film will not be obviously changed, and the concentrated Se can also stay in a molten state.

At the 350 °C, we have optimized the preheating duration (t_0). Because this is an initial parameter, we had only tried three conditions, that is, 150, 300 and 450 s. For the 150 s condition, only a little liquid Se is introduced on the precursor film, thus a relatively low average PCE of about 9.5% was obtained. 300 s is the optimal one among these three conditions. When the t_0 was increased to 450 s, much excessive Se was deposited on the precursor film, which gives a lower PCE of about 11%. From these data trend between 300 and 450 conditions, we can speculate that the t_0 could have an acceptable window of tens of second. This window should be easily controlled in experiment.

Subsequently, we have optimized the declining rate of the Se source temperature by controlling the annealing-stop temperature (T_E) ranging from 500 to 380 °C. As shown below, average cell efficiency of >11% can be obtained in a very wide parameter range from 440 to 380 °C. Average cell efficiency of >12% can be obtained in a range width of 40 °C, from 440 to 410 °C. In experiment, this large parameter window is highly acceptable and can be easily controlled.

Finally, we also slightly optimized the 540 °C selenization duration with another three parameters as the temperature profile shown below. In last two conditions, average cell efficiency of >11.5% can also be obtained.

Overall, the selenization route presented in this manuscript has a relatively large experiment window, which is easily controlled in experiment. More importantly, this route provides a large window (opportunity) for more precisely control the selenization processes.

2. In addition to the review comment #1, for the thermal cycle of selenization process, duration time at highest temperature and cooling temperature profiles for Se-source zone and CZTS precursor zone (after ~ 3500 s) should be also clearly displayed and discussed.

Reply: Thanks very much for this suggestion. The detailed temperature profile has been revised and given in Figure S12. After 3500 s, the heating is stopped, and the system cools naturally. We keep the lab room temperature at 25 °C, and the cooling duration is about 150 min.

Figure S12.

3. In Figure 1b, schematic of quartz furnace is shown. Can you add diameter of quartz diameter, the distance between Se-source (in Zone 1) and precursor film (in Zone 2)? Further, can you provide the distance between gas inlet and Se-source and the distance between precursor film and gas outlet. You also add some discussions how much margins are allowed in the distance for the high PCE? Much are not clear about your SLSG process.

Reply: Thanks very much for this question. For a better understanding of this selenization system, we have presented a schematic diagram in Figure S2, which gives almost all the needed information.

Figure S2

4. In the two zone furnace, 26 g of Se is placed both at Zone-1 and Zone-2. The Se amount affects the Se vapor pressure near precursor film. Can you also add acceptable Se-amount for the higher PCE? Se-partial pressure is important to obtain high PCE cell. As in your paper (Control of the phase evolution of kesterite by tuning of the selenium partial pressure for solar cells with 13.8% certified efficiency, Nature Energy), over- pressure (such as 1.6 atm) enables PCE more than 13 %. Can you add more information about the Se-vapor pressure too?

Reply: Thanks very much for this question. We need to note that all the Se is placed in zone 1. The only demand on the Se amount is that the quartz boats are always covered by liquid Se in the whole selenization process. The Se vapor generation rate is mainly determined by surface area of the liquid Se and the temperature, both of which can be easily controlled in experiment. In fact, we have not carefully optimized the Se amount used in the selenization. In this work, the selenization is performed under ambient pressure, that is, 1.0 atm. We have added this information in the experimental section for clarity. In the paper (Nature Energy 2023, 8, 526–535), The high pressure can help reduce the formation dynamics of binary and ternary secondary phases while in this work we use the liquid Se to drive the direct and fast formation of the CZTSSe phase.

5. In Figure 2, Raman data Cu_xSe and Cu_2SnSe_3 were used for fitting, In Figure S4, Se was used for Raman fitting. Can you provide XRD data for supporting the presence of those secondary phases? In the XRD data of Fig. S5, peaks are not assigned.

Reply: Thanks very much for this question. In the revised figure of the XRD data, the peaks

are assigned as below. The Se and Cu_xSe phase can be observed from the XRD. Cu_2SnSe_3 usually has the same XRD patterns with the CZTSSe and thus can hardly be distinguished. Raman spectra is the most effective method to distinguish the CTSe phase as we show in the manuscript.

Figure S7. XRD patterns of SG and SLSG samples in different intermediate stages: (a) 400 °C and (b) 500 °C. Three characteristics XRD peaks of Se_x can be clearly seen in the SLSG sample ($\sim 23.6^\circ$, 29.8° , 43.7° , referring to JADE PDF#06-0362). In the SG-400 °C sample, a small XRD peak corresponding to the Cu_{2-x}Se phase can be seen at about 44.3° (JADE PDF#06-0680), which is consistent with the Raman result. In addition, the SLSG sample shows more obvious CZTSSe XRD peaks, indicating that the crystallization of CZTSSe has been facilitated.

6. Generally, in chalcogenide, Cu_xSe is regarded as a flux to increase the grain size. You addressed the liquid Se assisted selenization of CZTSSe (as in the Figure S7). Can you discuss more about the difference between liquid Se-assisted growth and liquid Cu_xSe -assisted growth?

Reply: Thanks very much for this question. Indeed, as you said, Cu element has been found to be able to assist the grain growth. This arises mainly because (1) Cu has a higher reaction activity with the chalcogens to form $\text{Cu}_x(\text{Se}, \text{S})$ crystals and (2) that Cu atom has a higher mass transport ability in the solid phase. In addition, the $\text{Cu}_x(\text{Se}, \text{S})$ crystals also have a high reaction activity with other metal elements, facilitating the formation of ternary sulfides.

However, when referring to the Cu-Se phase diagram shown above (Inorganic Materials 36, 641–652 (2000)), we found that the Cu_xSe compound itself cannot form liquid phase to assist the grain growth at the typical selenization temperature. Although in the nano-micro scale in the polycrystalline film, the melting of Cu_xSe may be easier due to the surface tension, we still think the melting degree and the effect of crystal-assisted growth are not as good as that in a pure liquid phase. From the phase diagram, we can find that the formation of liquid phase at relatively low temperatures (<800 K) is mainly come from the molten Se and its induced eutectic $CuSe_2$ -Se. That is, to form liquid assisted crystal growth, sufficient Se is needed. As such, although the excessive Cu can also assist the crystal growth, its effect may be not as good as that of the liquid Se. In addition, the CZTSSe precursor film is usually in a Cu-deficient condition, which also limits the assistance effect of Cu_xSe phase.

7. In this manuscript, some FE-SEM images are presented. However, no EDS data are shown. Raman and XRD are presented for the phase evolution. Along with the data of the structural analysis, chemical compositions should be provided with EDS result. EDS result provide information of how much Se covers the films and how much Sn loss occurs at each selenization stages.

Reply: Thanks very much for this suggestion. To more clearly show the phase distribution and evolution, we have added EDX characterization of these samples. In the SG-400 sample, the metal elements exhibit homogeneous distribution, and no accumulation of Cu and Se has been

observed because this low temperature and short annealing duration can hardly drive significant atom/cation diffusion and accumulation. Considering the observed Cu_xSe phase in Raman and XRD, we think the formed Cu_xSe phase is in a form of nanocrystals. In SLSG-400 °C sample, the coverage of Se on the SLSG sample can be obviously observed. At a higher temperature of 540 °C, an enrichment of Se on the film surface can also be seen, which further supports the existence of liquid Se in the selenization process.

We think these added EDX results can better help us to understand the element distribution and phase evolutions. For the Sn loss, the EDX of SG-540 C-200s obtains $\text{Sn}/(\text{Cu}+\text{Zn}+\text{Sn})=0.275$, which is obviously lower than that of the SLSG sample (0.295). Together with the XRF results, the Sn loss can be confirmed.

8. Authors claimed 13.6 % for small cell and 12 % for large cell (over 1cm²). However, in the

certificated record in the supporting information, only PCE of small cell was displayed. Best PCE of small cell was shown as 13.45 % not 13.6 % in the certification. 13.45 % is better to be claimed. For the large cell PCE, it should be also presented by official certification with exact cell area and a photo of measured device.

Reply: Thanks very much for this suggestion. We have added the certification information in the revised version. Regarding the large-area device, we have made a certification during the manuscript reviewing process and it is added in Figure S20. The photo of the measured device is also given in the report.

Figure S20

Reply to Reviewer #3

Firstly, we would like to show our sincere appreciation to your kind reviewing, professional comments and suggestions. As you suggested, this manuscript has been thoroughly revised and detailed questions you proposed will be answered one by one in the following.

1. in the Methods section, it is stated that AgCl was added along with the other reactants. If that is true, then they do not have CZTSSe but $(\text{Ag}, \text{Cu})_2\text{ZnSn}(\text{S}, \text{Se})_4$ i.e. ACZTSSe. If this is correct then the authors should not only change the title but state it upfront in the manuscript. Furthermore, there are some early papers published on alloying with Ag that show improvement in carrier lifetime etc. and authors should cite pertinent papers from the literature. The efficiency results even after alloying with silver are still interesting but warrant a different discussion. (For example, rather than attributing the improvement entirely to selenization condition, it will be fair to say that improvement may be in addition due to alloying with silver).

Reply: Thanks very much for your comment. We have used Ag alloyed CZTSSe (ACZTSSe) to describe the composition of the Kesterite absorber in the manuscript including the experimental section. We also added five references (4-6, 40, 41) of the ACZTSSe absorber. A sentence describing the ACZTSSe has been added in manuscript, as

“In our experiment, 10% Ag alloying was used in the precursor, also to improve the absorber quality according to previous literatures (the final Ag-alloyed CZTSSe is abbreviated as ACZTSSe).^{4-6,40,41}”

Regarding the Ag alloying, we agree with you that it is very valuable for obtaining high-quality Kesterite film, by improving carrier lifetime and promoting the crystallization process. It is also an important topic in this field to further understand the mechanism of the effect of Ag alloying on the film fabrication, no matter for the conventional graphite system or for the two-zone system used in this work. Nonetheless, this deeper understanding of Ag alloying is beyond the topic of this manuscript. In the future, we will pay more efforts to clarify the important mechanism of cations alloying and we also hope to cooperate with more colleagues in this field to study this very interesting topic.

2. The details of the calculation profile in Figure S1 are needed. What are the pertinent equations? The details in Table S1 are not sufficient. It is not clear why the diffusion coefficient of water vapor in air and water density are listed in Table S1. By the same token, the diffusion coefficient for selenium is missing that must have been used in the simulation. Points A, B, C in Figure S1 really do not have sensors, do they? Currently, it implies that Se concentration at those points was measured. Furthermore, point A, must have been assumed to be in equilibrium with the Se droplet, if yes, then why is Se concentration rising with time at point A (is this due to changing temperature of the Se droplet)? All this confusion arises because boundary conditions in the simulation are not explicitly provided along with the equations.

Reply: Thanks very much for this question. We have added more simulation details in the SI (note 1) and we have also revised the initial mistakes in Table S1. The equations used in this simulation is relatively complicated and our simulation was performed by COMSOL.

Point A, B and C represent different positions in the simulated graphite box, at which the Se concentration evolutions were simulated. It is not a real measurement in experiment. Point A represents the position close to the Se droplet surface, which has not obtained thermodynamics equilibrium with the liquid Se droplet.

3. We need more information regarding their equipment setup. How long is their furnace tube? What was the distance between Zone 1 and Zone 2 in Figure 1 (b)? Is the precursor film along with its substrate put on top of another glass (the blue item shown in the Figure)?

Reply: Thanks very much for this comment. For clarity, we have added a schematic diagram to show the detailed experiment setup parameters in Figure S2. The circular quartz tube is a fixed part of the furnace, and the rectangular quartz tube is designed to obtain a smaller reaction space and higher operating convenience (much easier for Se source and sample placing).

The precursor film along with its substrate is put on a graphite plate to obtain uniform heating.

Figure S2

4. t_p in Figure 1 (c) should be t_0 .

Reply: Thanks very much for this comment. We have modified to show t_0 more clearly.

5. The authors talk about all the grain growth etc. in detail with reference to SG and SLSG. However, SLSG gave poor cell performance and they shifted to SLSG-O. However, very little detail is provided for SLSG-O. Are they referring to decreasing the temperature of the Se source around 2000 S (I am assuming it to be in seconds – they should list proper units for both the X and Y axis in this figure) as shown in Figure 1(c)? If yes, then is the figure drawn to scale? Specifically, they should tell us how fast the Se source temperature decreased (the initial and final temperatures with the time duration).

Reply: Thanks very much for this question. We are sorry we did not clearly show the temperature profile in the initial manuscript. Honestly, Figure 1(c) is a schematic diagram and in the revised version we have added more exact temperature profile in Figure S3 and S12.

Figure 1(c)

Figure S12

6. An important item missing is the final cooling step. After the completion of selenization at 540C, how were the sample and the Se source cooled? This profile is missing in the text and is very much needed.

Reply: Thanks very much for this question. After the 540 °C selenization, the heating of these two zones is stopped at the same time and these two zones are cooled naturally. The room temperature of our lab is kept at 25 °C to obtain a relatively stable natural cooling rate. A detailed temperature profile of this work is given in Figure S12.

7. They should tell us how the blue region in Figure 2(a2) was deduced to be cooled liquid Se.

Reply: Thanks very much for this comment. We have revised this figure, and the cooled liquid Se is determined from the EDX mapping. As shown in the below figure, in the SLSG sample, high Se EDX intensity can be seen in the left side of the film while all the metal elements show relatively lower EDX intensity in this region. In contrast, in the SG sample, the Se exhibits a synchronous distribution with the metal elements. These results indicate that elemental Se exists on the surface of the SLSG sample (left side). We have revised Figure 2(a-b) with these new SEM figures, and the EDX mapping is given in the Figure S5. Compared to the metal sulfide crystals, the cooled Se exhibits an amorphous morphology without clear shape.

Figure S5.

8. In Figure 3f, how many solar cells were studied? The deviation is over how many samples? Do the samples include cells from different batches or are they from the same overall film (i.e. six cells from their one coated film shown in SI)? This is important to understand reproducibility.

Reply: Thanks very much for this question. In the statistical analysis, 18 cells have been involved (three to four pieces of coated films). These cells were fabricated in two to three selenization batches. In Figure 3f, the performance data has been added in the box graph for clarity and the detailed parameters have been given in Table S2.

9. It seems that their reported efficiencies are based on the total cell area. It will be good to state that once in the main text.

Reply: Thanks very much for this suggestion. We have added this statement in the abstract and the introduction sections.

10. On page 13, they state ‘while the rise of TPC signal of the SLSG-O sample is obviously slowed down ...’. It is not obvious at all. They need to explain to clarify for the reader what they mean by it.

Reply: Thanks very much for this comment. We are sorry we made a mistake in describing the TPC behavior in the initial manuscript. We have revised this part as below:

Additionally, the time position of the photocurrent peak for the SLSG-O sample remained almost constant regardless of the applied voltage, while that of the SG sample notably shifted with increasing voltage. This indicates that the SLSG-O sample possesses a more stable and voltage-independent charge transport ability, primarily benefiting from improved crystallization quality, reduced presence of secondary phases, and diminished carrier trapping states.

11. For the large area device, they need to show its picture. How are the grids patterned? It will help in assessing that indeed current was collected from the large area.

Reply: Thanks very much for this suggestion. The photograph of the large-area device (after certification) is added in the certification report in Figure S20. As shown below, in one piece of substrate, two large-area cells are fabricated, and they are separated by mechanical scribing. The area for measurement is determined by the shading mask. We need to note that the grid electrode patterns need to be further optimized to improve the cell FF.

Figure S20.

REVIEWER COMMENTS

Reviewer #1 (Remarks to the Author):

They provided detailed answers to the questions and made careful revisions to their manuscript based on the reviewers' feedback.

I was able to understand the analysis and descriptions of their data, especially the film growth parts.

Therefore, I believe that this revised manuscript is suitable for publication in this journal.

Reviewer #2 (Remarks to the Author):

See attachment.

Dear Editor,

We found that authors revised paper well according to the reviewer's comments. However, to be accepted some minor revisions are required. Following points should be clarified and addressed.

1. On the fabrication method, absorber precursor was spin-coated on pre-cleaned Mo substrate. Can you describe the pre-cleaning method for Mo substrate?
2. For the fabrication of the absorber films, spin-coating and annealing were repeated. Can you indicate where the spin-coating and annealing process were done, in glove box or in an air?
3. i-ZnO and ITO were deposited by sputtering method. Can you add the more information of the sputtering; sputtering power, operation gas and operation pressure? For the high PCE devices, sputtering damage may be also considered.
4. For the analysis of Figure 4 (a)~ 4(e), more explanation and discussion should be provided. Those results are not obtained by commercial equipment, but by lab-made set up. Thus, it needs more description of set-up and of the way that the results are interpreted.
5. For comparison, can you add cross-sectional FE-SEM images of completed devices? We want to know how much grain size is affected by the liquid Se assisted growth. In addition, for further comparison we need a FE-SEM image of completed device made by using single graphite box.
6. On the previous comment, following is responded:

“Generally, in chalcogenide, Cu_xSe is regarded as a flux to increase the grain size. You addressed the liquid Se assisted selenization of CZTSSe (as in the Figure S7). Can you discuss more about the difference between liquid Se-assisted growth and liquid Cu_xSe -assisted growth?

Reply: Thanks very much for this question. Indeed, as you said, Cu element has been found to be able to assist the grain growth. This arises mainly because (1) Cu has a higher reaction activity with the chalcogens to form $Cu_x(Se, S)$ crystals and (2) that Cu atom has a higher mass transport ability in the solid phase. In addition, the $Cu_x(Se, S)$ crystals also have a high reaction activity with other metal elements, facilitating the formation of ternary sulfides.

7.

However, when referring to the Cu-Se phase diagram shown above (Inorganic Materials 36, 641–652 (2000)), we found that the Cu_xSe compound itself cannot form liquid phase to assist the grain growth at the typical selenization temperature. Although in the nano-micro scale in the polycrystalline film, the melting of Cu_xSe may be easier due to the surface tension, we still think the melting degree and the effect of crystal-assisted growth are not as good as that in a pure liquid phase. From the phase diagram, we can find that the formation of liquid phase at relatively low temperatures ($<800\text{ K}$) is mainly come from the molten Se and its induced eutectic $\text{CuSe}_2\text{-Se}$. That is, to form liquid assisted crystal growth, sufficient Se is needed.

As such, although the excessive Cu can also assist the crystal growth, its effect may be not as good as that of the liquid Se. In addition, the CZTSSe precursor film is usually in a Cu-deficient condition, which also limits the assistance effect of Cu_xSe phase.

Overall, authors' response is acceptable. If Se is supplied enough or much (, which is generally required for selenization process), the crystallization is enhanced (, which is also together with Cu-Se phase). In the higher Se concentration, CuSe_2 and liquid phase are observed in the phase diagram. As Cu-Se is easily formed at a lower temperature and Cu has higher mobility, the Cu-Se phase plays a main role in increasing grain size and in enhancing formation-reaction of kesterite crystal structure. For CIGS and CZTSSe, if we vary the Cu-concentration from Cu-poor to Cu-rich (for example, Cu ratio $(\text{Cu}/(\text{In}+\text{Ga}), \text{Cu}/(\text{Zn}+\text{Sn}))$ from 0.6 to 1.2), the grain size is observed

to be increased with high Cu-concentration ratio. This trend is direct indication of the role of Cu-Se associated with grain growth. When selenization is conducted with single graphite box, where liquid Se effect as shown in this manuscript is not associated, grain growth can be enhanced (irrespective of Cu-concentration ratio if Se-vapor partial pressure is enough), which is due to the enhanced formation of Cu-Se phase. Thus, the role of Cu-Se during the selenization process is not negligible even in Cu-poor CZTSSe, and we recommend that authors add some additional comments on the role of Cu-Se in addition to the liquid Se assisted growth.

Reply to Reviewer #2

Firstly, we would like to show our sincere appreciation to your kind reviewing, professional comments and suggestions. As you suggested, this manuscript has been thoroughly revised and detailed questions you proposed will be answered one by one in the following.

1. On the fabrication method, absorber precursor was spin-coated on pre-cleaned Mo substrate. Can you describe the pre-cleaning method for Mo substrate?

Reply: Thanks very much for this comment. We have added the description of the pre-cleaning process of the Mo substrate in the experimental section as “The Mo substrate was pre-cleaned using dish soap and deionized water (including 1 min ultrasonic cleaning in water) and was dried by N₂ gas flow.”

2. For the fabrication of the absorber films, spin-coating and annealing were repeated. Can you indicate where the spin-coating and annealing process were done, in glove box or in an air?

Reply: Thanks very much for this comment. We have added the experimental condition for the spin coating process in the experimental section as “This coating-annealing process was repeated 4 times in ambient conditions (25 °C, uncontrolled humidity, air).”

3. i-ZnO and ITO were deposited by sputtering method. Can you add the more information of the sputtering; sputtering power, operation gas and operation pressure? For the high PCE devices, sputtering damage may be also considered.

Reply: Thanks very much for this comment. We have added the experimental condition for the sputtering process in the experimental section as “followed by sputtering 50 nm i-ZnO layer (sputtering power: 55 W, gas source: Ar, pressure: ~2 Pa, temperature: 60 °C) and 200 nm ITO layer (sputtering power: 60 W, gas source: Ar, pressure: ~0.2 Pa, temperature: 160 °C).”

We greatly agree with you that the sputtering damage could influence the performance of CZTSSe solar cells. We are also exploring alternative method to minimum this influence (such as ALD or solution processed ZnO as that has been done in CIGS solar cells);

however, no obviously better result than the sputtered ZnO has been obtained until now. Nonetheless, it is an important point for fabricating more efficient CZTSSe solar cells.

4. For the analysis of Figure 4 (a)~ 4(e), more explanation and discussion should be provided. Those results are not obtained by commercial equipment, but by lab-made set up. Thus, it needs more description of set-up and of the way that the results are interpreted.

Reply: Thanks very much for this comment. We have added more description of this measurement and analysis in the main text of the manuscript (page 15-16 in red).

Including “This method can probe the charge transport and recombination dynamics behaviors of a completed cell and these charge dynamics properties can further be used to quantify the bulk and interface charge loss in the cell. The detailed experimental setup and analysis model have been described in our previous works.⁵¹⁻⁵²”

“Quantifying the modulated electrical transient measurements demonstrated a higher charge collection (η_C) efficiency in the SLSG-O sample (Figure 4(e)),⁵² confirming that the improvement in the heterojunction properties has reduced interface charge loss in the cell.”

“This reduction in bulk defect density agrees with the improved charge transport observed from the M-TPC, which resulted in a significant improvement in the charge extraction efficiency (η_e) of the SLSG-O cell, particularly in the high-voltage regime (Figure 4(e)). The enhancement in the η_e indicated that charge loss in the bulk absorber has been reduced, which primarily benefited from both the lower defect density and lower charge capturing velocity due to improved ACZTSSe lattice ordering.”

5. For comparison, can you add cross-sectional FE-SEM images of completed devices? We want to know how much grain size is affected by the liquid Se assisted growth. In addition, for further comparison we need a FE-SEM image of completed device made by using single graphite box.

Reply: Thanks very much for this comment. We have added the cross-sectional SEM images of the completed cells (Mo/absorber/CdS/ZnO/ITO) with the absorber fabricated by different methods in Figure S19. It is obvious that the SLSG-O sample has the best cross-sectional morphology with larger grains in the fine-grain layer.

Figure S19. Cross-sectional SEM images of the completed cells with the Kesterite absorber fabricated by different methods. It can be seen that the SLSG-O sample has the best cross-sectional morphology, especially the fine grain layer has much larger grain size.

6. Overall, authors' response is acceptable. If Se is supplied enough or much (, which is generally required for selenization process), the crystallization is enhanced (, which is also together with Cu-Se phase). In the higher Se concentration, CuSe₂ and liquid phase are observed in the phase diagram. As Cu-Se is easily formed at a lower temperature and Cu has higher mobility, the Cu-Se phase plays a main role in increasing grain size and in enhancing formation-reaction of kesterite crystal structure. For CIGS and CZTSSe, if we vary the Cu-concentration from Cu-poor to Cu-rich (for example, Cu ratio (Cu/(In+Ga), Cu/(Zn+Sn)) from 0.6 to 1.2), the grain size is observed to be increased with high Cu-concentration ratio. This trend is direct indication of the role of Cu-Se associated with grain growth. When selenization is conducted with single graphite box, where liquid Se effect as shown in this manuscript is not associated, grain growth can be enhanced (irrespective of Cu-concentration ratio if Se-vapor partial pressure is enough), which is due to the enhanced formation of Cu-Se phase. Thus, the role of Cu-Se during the selenization process is not negligible even in Cu-poor CZTSSe, and we recommend that authors add some additional comments on the role of Cu-Se in addition to the liquid Se assisted growth.

Reply: Thanks very much for this comment. We agree you that the Cu-Se phase could assist the crystal growth and have added more description about this positive effect as “**In addition to the Se itself, it was reported that Cu-Se compound could also perform as flux to assist the growth of chalcogenides due to its high reaction activity with other metal elements and the high mobility of Cu cations. In our samples, high Se concentration would facilitate the**

melting of Cu-Se alloying to more effectively assist the crystal growth.”

REVIEWERS' COMMENTS

Reviewer #2 (Remarks to the Author):

Dear Editor,

I found that authors revised manuscript well in response to the reviewer's comments. I am pleased to recommend the manuscript for publication in Nature Communications.

Thank you.